# Nanoliposomes in Cancer Therapy: Marketed Products and Current Clinical Trials

**DOI:** 10.3390/ijms23084249

**Published:** 2022-04-12

**Authors:** Raquel Taléns-Visconti, Octavio Díez-Sales, Jesus Vicente de Julián-Ortiz, Amparo Nácher

**Affiliations:** 1Department of Pharmacy and Pharmaceutical Technology and Parasitology, Pharmacy Faculty, University of Valencia, 46100 Valencia, Spain; raquel.talens@uv.es (R.T.-V.); octavio.diez@uv.es (O.D.-S.); amparo.nacher@uv.es (A.N.); 2Instituto Interuniversitario de Investigación de Reconocimiento Molecular y Desarrollo Tecnológico (IDM), Universitat Politècnica de València, Av. Vicent Andrés Estellés s/n, Burjassot, 46100 Valencia, Spain; 3Molecular Topology and Drug Design Research Unit, Department of Physical Chemistry, Pharmacy Faculty, University of Valencia, 46100 Valencia, Spain

**Keywords:** nanoliposomes, cancer, translational medicine, clinical trials

## Abstract

The drugs used for cancer treatment have many drawbacks, as they damage both tumor and healthy cells and, in addition, they tend to be poorly soluble drugs. Their transport in nanoparticles can solve these problems as these can release the drug into tumor tissues, as well as improve their solubility, bioavailability, and efficacy, reducing their adverse effects. This article focuses on the advantages that nanotechnology can bring to medicine, with special emphasis on nanoliposomes. For this, a review has been made of the nanoliposomal systems marketed for the treatment of cancer, as well as those that are in the research phase, highlighting the clinical trials being carried out. All marketed liposomes studied are intravenously administered, showing a reduced intensity of side-effects compared with the nonliposomal form. Doxorubicin is the active ingredient most frequently employed. Ongoing clinical trials expand the availability of liposomal medicines with new clinical indications. In conclusion, the introduction of drugs in nanoliposomes means an improvement in their efficacy and the quality of life of patients. The future focus of research could be directed to develop multifunctional targeted nanoliposomes using new anticancer drugs, different types of existing drugs, or new standardized methodologies easily translated into industrial scale.

## 1. Introduction

Nowadays, cancer remains one of the leading causes of morbidity and mortality in the world, with approximately 19.3 million new cases worldwide in 2020 according to the latest data available worldwide estimated within the GLOBLOCAN project [1]. The most commonly diagnosed cancers worldwide in 2020 were breast (2,261,419), trachea, bronchus and lung (2,206,771), colorectal (1,931,590), prostate (1,414,259), nonmelanoma skin cancer (1,198,073), and stomach (1,089,103), all of which accounted for more than one million new cases. Population estimates indicate that the number of new cases will increase in the next two decades to reach 29.5 million in 2040. The estimated number of new cases worldwide from 2020 to 2040 is shown in Figure 1. According to this estimate, the percentage change will be more than 20% for all cancer types (+83.6% for nonmelanoma skin cancer, +72.8% for bladder, +71.6% for prostate, +70.2% for pancreatic, +66.9% for colon, +64.4% trachea, bronchus, and lung, etc.).

Against this backdrop, new technologies have emerged to help alleviate the situation. Among them, nanotechnology stands out.

Nanomaterials have been the subject of great interest by many researchers in recent decades due to their potential role in improving public health [2]. In fact, research into biomedical applications for nanoparticles has led to the development of various medical and pharmaceutical devices, such as drug carriers and biosensors.

Nanomaterials are structures with specific size-dependent properties [3]. Nanometric materials usually measure up to 100 nm, although, for example, the European Medicines Agency (EMA) considers structures up to 1000 nm [4]. Nanometric-sized particles show different physicochemical characteristics compared to larger particles, which gives them novel interactions in the biological environment [5]. For this reason, in recent years, the use of these materials in the diagnosis, prevention, and treatment of different diseases has been explored with great interest. It is not new that conventional medicine offers serious drawbacks in the treatment of some disorders and needs to be replaced.

Nanomaterials show a large surface/volume ratio, which facilitates their simultaneous interaction with a large number of molecules. Thus, with the use of nanoparticles, the apparent solubility of lipophilic substances can be increased, and the stability of different drugs can be improved. In addition, nanosized materials can control drug release and selectively target molecules to a specific site through the selective effect of permeation and retention, or by functionalization with receptor-specific ligands, such as peptides, antibodies, and small molecules [6,7]. In addition, this functionalization of the surface of the nanoparticles, the modification of the surface charge, and the modulation of its size allow the therapeutic action to be directed to specific tissues and also to modify different pharmacokinetic parameters, such as half-life and bioavailability. These characteristics play a key role in the design of new theranostic therapies or drugs [8,9,10,11].

Nanomaterials can be classified into organic and inorganic nanomaterials according to their chemical composition [12]. Hybrid nanomaterials have also been developed in response to the demand for highly functionalized materials. The characteristic of organic nanomaterials, mainly composed of natural or synthetic polymers and lipids, includes good biocompatibility, high biodegradability, and low toxicity [2]. Examples of these are polymeric nanoparticles, lipid nanoparticles, nanofibers, nanoemulsions, liposomes, and carbon nanotubes [13].

When a drug, or biologically active molecule, is administered in a conventional way (tablet, injection, etc.), the drug is quickly released into the biological environment and its behavior in the body depends entirely on its chemical structure. In reality, the chemical structure determines the physicochemical properties of the active molecule (molecular weight, lipophilicity, ionization, etc.), which determine its eventual absorption through the membranes and its distribution in the body. When distributed in the body, the drug molecules can reach the diseased tissue but, at the same time, reach healthy regions or tissues, inducing side- or undesirable effects [14]. On the contrary, when the active molecule is administered in the form of nanoparticle, it is its physicochemical properties (size, surface characteristics) that determine the distribution of a drug in the body and its concentration at the site of action. Once the tissue where they must perform their function has been reached, the drug-carrying nanoparticles must control the release of their contents. The fact of being able to concentrate the drug in its place of action or absorption offers the possibility of minimizing side-effects and increasing its therapeutic index. This approach is of vital importance in the development of new drugs for the treatment of cancer, as well as in the improvement of current treatments or to administer certain antitumor drugs [15].

Complexes of new nanocarriers with drugs can effectively remove molecular constraints on their entry into the body [16]. Through the use of nanocarriers such as nanoliposomes, cytostatic drugs can be seamlessly loaded and inserted into the appropriate tissues in the body [17,18,19]. On the one hand, nanotechnology increases the solubility of poorly soluble compounds and improves its stability. Moreover, nanotechnology can be applied to improve the efficacy of the therapeutic drug and reduce its side-effects [20]. Probably one of the most widespread applications in the field of therapy is the use of nanoparticles as transport vehicles for the controlled release of drugs [21].

Another interesting field where nanocarriers can have a clinical application is as systems for the administration of molecules of biotechnological origin, including peptides, proteins, antisense oligonucleotides, and plasmids. These active molecules are very sensitive to physical-chemical and enzymatic degradation, they are unable to cross biological barriers (mucosa) and, in addition, in some cases, they must reach very precise cellular compartments to be able to exert their action. For this reason, its inclusion in nanoparticles allows the solution of part or all of these problems, offering interesting possibilities for their administration in a safe and effective way [22].

## 2. Nanoliposomes

Liposomes were discovered by Bangham in the 1960s when observing that some lipids form membranous structures when they come into contact with an aqueous medium [23,24]. The vesicles generated are round in shape and are composed of one or more phospholipid bilayers surrounding an aqueous core; this feature allows the encapsulation of both hydrophilic and lipophilic active substances [25,26]. Phospholipids with or without the incorporation of cholesterol, surfactants, and other materials are used for their preparation. They are incorporated into the phospholipid bilayer to modify some of the properties of the biomembrane: surface charge, membrane permeability, or lipid stability in the bilayer [27,28]. In this sense, the presence of cholesterol enhances the mechanical strength of membranes, influencing membrane elasticity and increasing the packing density of lipids as a function of the proportion included. Moreover, with the incorporation of cholesterol, the liposomes prepared are expected to be more stable in the presence of biological fluids [29]. Due to this, cholesterol is often incorporated in the formulation of liposomes used in drug delivery [30]. The phospholipids that make up liposomes can have saturated or unsaturated hydrocarbon chains and show different degrees of stiffness and permeability. Thus, the compositions that contain a higher proportion of saturated phospholipids are more rigid and impermeable, compared to the bilayers consisting of unsaturated phospholipids [31]. The phospholipids used in the manufacture of liposomes are commonly natural or synthetic phosphatidylcholines (dimyristoyl, dipalmitoyl, or distearoylphosphatidylcholine). Among them, anionic phospholipids (phosphatidic acid or phosphatidylserine) or cationic phospholipids (stearylamine) can be used, providing liposomes a specific net surface charge [32]. The liposomal surface charge determines the interaction of liposomes with tissues and cells, which can affect the safety issues [33,34]. Some negative effects of liposomes, especially cationic liposomes, have been observed [11,35,36]. Modification of the liposome surface to improve its properties, such as immunocompatibility, for example, with a polyethylene glycol coating, has also opened new horizons [37].

In the 1970s, Gregoriadis began studying these liposomes as delivery systems for drugs or other bioactive molecules. The advantage of this vehicle is based on the ability to incorporate hydrophilic, hydrophobic, and amphiphilic molecules [38,39,40]. As mentioned, these delivery systems have been developed to modify the route of administration, improve bioavailability, and change the release profile or improve a formulation. To obtain liposomes of different sizes, the use of diverse preparation methods is required [32,41]. The simplest and most widely used is hydration of the lipid film with an aqueous buffer at a temperature above the lipid transition temperature. The drug to be encapsulated can be included in the aqueous core, in the case of hydrophilic drugs, or in the lipid film, in the case of lipophilic drugs. This preparation method forms multilamellar vesicle-like lipids and, by extrusion, liposomes can be obtained from a single lipid bilayer. A disadvantage of this method is its low encapsulation efficiency of hydrophilic drugs. There are other methods of preparing liposomes [32].

In recent years, considerable progress has been seen in the investigation of liposomes as transport vesicles of therapeutic agents. It can be said that liposomes have some indubitable advantages such as improving the efficacy and therapeutic index of the drugs, and their stability by encapsulation. Furthermore, the delivery of the drug can be controlled in response to an internal (pH, presence of enzymes, or a change in redox potential) or external (light, temperature, magnetic field) stimulus [42], and that can simultaneously transport several drugs and/or biomacromolecules (peptides, proteins, and nucleic acids). They are nontoxic, flexible, biocompatible, fully degradable, and nonimmunogenic. They reduce the toxicity of encapsulated agents (Taxol, Amphotericin B, etc.) and help reduce the exposure of tissues sensitive to toxic drugs [43,44]. 

Importantly, liposomes have certain limitations, such as the inability to remain in the systemic circulation for prolonged periods. Conventional liposomes administered intravenously are eliminated by the endothelial reticulum system, when they are detected as foreign materials to the body, with the help of certain host molecules such as opsonins.

Different researchers have studied liposomes coated with hydrophilic polymers (PEG), and other polymers such as polyacrylamide, polyvinyl alcohol, or polyvinylpyrrolidone have been used to achieve the same goals. These pegylated liposomes are not detected by the immune system and can access their place of action, improving their efficacy and increasing the bioavailability of different drugs [45]. PEG is a linear or branched polyether with a hydroxyl group at each end; it is highly soluble in water as well as in various organic solvents and is FDA-approved for administration in humans. It is inert, nontoxic, and nonimmunogenic, and it is easily discarded by the body through the kidney (molecular weight of the polymer less than 20 kDa), or from the liver (molecular weights greater than 20 kDa). It is incorporated on the surface of nanodrugs to create the so-called ‘steric stabilization’ effect, where PEG molecules form a protective hydrophilic layer on the surface of nanodrugs, preventing interaction with each other (aggregation) and with blood components. As a result, pegylation reduces macrophage uptake and prolongs blood circulation times [46].

Gene therapy consists of introducing nucleic acids (DNA or RNA) into the target cells to modify the genetic information or the expression of certain proteins at the cellular level. On this basis, gene therapy can be applied in various ways, being able to provide new functionality to cells, replace functions lost by the disease, or even inhibit the expression of certain genes, preventing the spread of cancer. The development of effective and safe gene-delivery systems is the main bottleneck toward successful clinical applications in different fields (e.g., cancer gene therapy). Nonviral gene-delivery systems hold the potential to bypass the limitations of their viral counterparts. Among these, liposomes emerged as a preferred platform (Figure 2) [47,48]. Thus, new multifunctional drug delivery systems may include components that allow the application of imaging and diagnostic techniques capable of identifying the stage of a certain pathology [42,49,50,51]. In fact, advances in nanoliposomes are facilitating the treatment of various diseases, such as cancer, neurodegenerative, autoimmune, and cardiovascular diseases. 

Nanoliposomes having a composition similar to that of living cells cause fewer problems than other nanoparticles do (made of polymers, silicon, and metals). However, recently, attention has been paid to several factors affecting the nanoliposomes’ safety, such as the size, composition, surface charge, stability, incorporation into tissues, and interaction with the cells, as they represent key factors in the medical use. In particular, although optimal liposome diameters could be between 100 and 200 nm for penetration into tumor cells as opposed to healthy tissues, apparent benefits have been observed for ultra-small liposomes, with diameters of 20–30 nm. With respect to surface charge, some adverse effects for cationic liposomes such as higher toxicity have been observed. Comparative studies have confirmed that used and approved nanoliposomal drugs practically show an extremely low frequency of side-effects, and the expanding trend for their intensive application is in the oncology field, for drug delivery and tumor cell imaging [52]. Within this context, the increase in the number of publications and research using nanoliposomes is a reflection of the advantages that nanotechnology can bring to medicine.

## 3. Application of Nanoliposomes in Cancer Therapy

Cancer continues to be one of the main causes of morbidity and mortality in the world. In principle, this disease is located in a certain tissue or organ and, later, it spreads to other distant places (metastasis), a process that involves a series of changes that make cancer a very complex disease [53]. Treatments available to treat cancer include surgery to remove the tumor, chemotherapy, radiation therapy, immunotherapy, and targeted and hormonal therapies. However, most patients receive a combination of surgery with chemotherapy or radiation therapy. Treatment falls largely on chemotherapy. In fact, the benefit of most conventional chemotherapeutics is limited either by the inability to deliver therapeutic drug concentrations to the target tissues or by severe and toxic effects on normal cells and tissues.

The mechanism of action of most chemotherapeutic agents relies on the interference of cell proliferation to prevent uncontrolled cell division of cancer cells. However, healthy cells are also in a constant proliferation process, so that an antineoplastic will not discriminate between a healthy and a malignant cell, resulting in very severe adverse effects during treatment.

One of the most important characteristics that distinguishes tumor tissue is that, for cells to grow rapidly and uncontrollably, there must be a stimulation of angiogenesis, which results in a faulty architecture forming a highly porous system (10 to 800 nm). This unique pathophysiology, in combination with poor lymphatic drainage, is known as the enhanced permeability and retention effect (EPR). Thus, nanosized systems reach the tumor tissue by passive diffusion, enter through the pores, and remain there for a long time due to poor lymphatic drainage. In this way, concentrations of this type of vehicle can be reached up to 10 times higher in the tumor compared to healthy tissue [53].

In addition, tumor cells often overexpress some types of membrane receptors that promote their uncontrolled growth. Thus, the nanosystem can be modified on the surface or decorated with a certain ligand (peptide, protein, or antibody) that recognizes the receptor, allowing preferential accumulation in tumor tissue [54,55,56]. Transporting anticancer drugs into nanoparticles allows us to solve these problems, among others, as they can release the drug that affects tumor tissues, as well as improve their solubility, their bioavailability, and their efficacy while reducing their adverse effects. In this line of action, cancer stem cells (CSCs) are a subject of intense research that aims at the development of efficient targeting therapies. CSCs have been identified in almost all types of cancers and can act as a reservoir of cancer cells that may cause a relapse after surgery, radiation, or chemotherapy. In fact, chemoresistance is a major problem in cancer therapy that contributes to poor prognosis and survival rates of patients. CSCs play a key role in this event due to their unlimited proliferative ability and multidrug resistance [57]. Thus, they are obvious targets in therapeutic approaches and also a great challenge in cancer treatment [58]. In this sense, liposome nanoparticles, with various sizes and modifications to their surfaces, can be easily prepared and offer promising means for targeting and eradicating CSCs for cancer therapies [59].

Nanoliposomes are currently considered to be the most successful drug delivery systems due to their biological and technological advantages [60]. In support of this, we can point out the following. If we search pubMed for ‘(liposome OR nano-liposome) AND cancer’, we obtain 17,491 results. Of these, 2957 are reviews and meta-analyses. Figure 3 shows the annual evolution of these publications until 2021.

Several liposomal drugs have been approved for clinical use. Table 1 shows liposome-based products marketed for cancer treatment.

**Marqibo** is the trade name for Vincristine encapsulated in a liposome, approved by the FDA in 2012, but not approved by the EMA. It is a Vinca alkaloid that is used as an anticancer agent because it acts on the metaphase of mitosis, binding to tubulin and preventing it from polymerizing, thus producing an alteration in the mitotic spindle, metaphase arrest, and apoptotic cell death. A short exposure to Vincristine makes this process reversible, but if exposure is prolonged, it can become lethal, so its antitumor capacity depends on its concentration, exposure time, and the number of cells that are undergoing mitosis while exposed to the drug [61]. Vincristine has been used for about 45 years for the treatment of leukemias, lymphomas, and solid tumors. However, it has a number of drawbacks that have been solved with the synthesis of Marqibo. Vincristine is poorly soluble in aqueous solutions at physiological pH, and has a large volume of distribution, rapid plasma clearance, and low elimination half-life. The liposomal form increases the half-life of Vincristine and its release into the tumor tissue, thus achieving higher drug concentrations in the tumor without increasing toxicity [61]. Marqibo has been investigated in several clinical trials to treat lymphoblastic leukemia, Hodking’s lymphoma, non-Hodking’s lymphoma, and solid tumors. Marqibo monotherapy was effective as third-, fourth-, and fifth-line monotherapies. It was approved by the FDA to treat lymphoblastic leukemias in patients whose disease progresses or has relapsed two or more times after anticancer therapy [61].

Different products for the same drug have been marketed, **Doxil** (USA) and **Caelyx** (Europe and Canada). Doxil is the first nanoparticle approved by the FDA, in 1995, while Caelyx was approved by the EMA later. It is a PEG-coated liposome containing Doxorubicin hydrochloride and is administered intravenously. Doxorubicin acts by inserting itself between the base pairs of the DNA chain, preventing its replication and, therefore, the multiplication of cells [62]. It is used for all types of tumors and is considered one of the first-line drugs for the treatment of cancer due to its great effectiveness. Doxil (Caelyx) is used specifically for the treatment of Kaposi’s sarcoma, multiple myeloma, ovarian cancer, and breast cancer [63]. Doxorubicin has the same adverse effects as the other anticancer drugs (nausea, vomiting, stomatitis, alopecia, myelosuppression, etc.), but the most toxic reaction is the cardiotoxicity it produces, causing irreversible congestive heart failure. This failure is dose-dependent, when the dose exceeds 450–550 mg/m^2^ [62]. Moreover, it is poorly soluble, has a high volume of distribution, and has a high systemic clearance rate (over 55 L/h). Both parameters are reduced by introducing it into a liposome [64].

The PEG coating of the liposome protects it from opsonization by plasma proteins, thus preventing it from being taken up by the mononuclear phagocytic system and circulating longer in the plasma [65]. However, this pegylated liposome of Doxorubicin has been shown to produce hand-foot syndrome. This is a syndrome that causes skin rashes on the soles of the feet and palms of the hands, causing treatment to be discontinued and the dose to be decreased [66].

**Myocet**, as with Doxil and Caelyx, is another liposome containing Doxorubicin. However, it does not contain PEG on its surface, so it does not produce hand-foot syndrome [66]. It is indicated for the treatment of metastatic breast cancer in combination with Cyclophosphamide. **Onivyde** is a pegylated liposome approved by the FDA in 2015 and by the EMA in 2016. It contains Irinotecan, a cytotoxic alkaloid that inhibits topoisomerase I, an enzyme involved in DNA replication and transcription. Nonliposomal irinotecan is used for colon and rectal cancer, while in metastatic pancreatic cancer, it has shown little efficacy when administered alone, but increases when administered together with folic acid, oxaliplatin, and 5-fluorouracil. Onivyde is administered intravenously in combination with 5-fluorouracil and folic acid as second-line treatments of metastatic pancreatic cancer, as is nonliposomal Irinotecan, in those patients who have progressed after being treated with Gemcitabine [67]. Among the disadvantages of nonliposomal Irinotecan are its toxicity and rapid elimination. Onivyde, on the other hand, increases the amount of Irinotecan that reaches the tumor tissue, as the liposome accumulates in it thanks to the EPR effect. Moreover, as it is pegylated, its half-life is longer than that of the nonliposomal form [67]. On the other hand, Onivyde has much smaller volume of distribution (2.61 L/m^2^) than nonliposomal Irinotecan (138 L/m^2^), so it is found in greater quantities in the intravascular fluid [68]. During treatment with Onivyde, there is no need to adjust the dose in people with mild or moderate renal insufficiency [67], and no cases of peripheral neuropathy have been reported. It has also been shown to increase overall survival, but only by approximately 2 months [69].

**Daunoxome** is a liposome containing Daunorubicin for the treatment of Kaposi’s sarcoma, acute myeloid leukaemia, and a wide variety of solid tumors. Daunorubicin is an anthracycline that causes cell apoptosis by different mechanisms of action: its metabolism gives rise to radicals that damage the DNA chain, it intercalates between DNA and RNA base pairs blocking replication, and it blocks topoisomerase II [70]. It is a liposome that is uncharged and has a very small size (45 nm), so it will be less taken up by the endoplasmatic reticulum system. The clearance is much lower in Daunoxone than in conventional Daunorubicin (10.5 mL/min vs. 233 mL/min, respectively) as well as having a longer half-life. This implies that doses of Daunorubicin can be increased inside the liposome without increasing toxicity [70]. The dose of Daunoxome is 40 mg/m^2^ every two weeks and is linked to a decrease in neutropenia, neuropathy, and alopecia and does not produce a decrease in cardiac function [71]. In clinical trials, Daunoxome was shown to generate a better response to treatment than bleomycin and vincristine, a combination therapy commonly used to treat Kaposi’s sarcoma [72]. In addition, Daunoxome has been shown to be very stable, to protect Daunorubicin against chemical and enzymatic degradation, and to decrease its binding to plasma proteins [71]. It also produces an increase in bioavailability and decreases metabolites formed. In addition, it favors a greater accumulation of the drug in tumor tissues rather than in healthy tissues.

Liposomal **Vyxeos** is a cancer medicine authorized by the EMA in 2018 and used to treat adults with newly diagnosed acute myeloid leukaemia. It is used when the leukaemia was caused by previous treatments (e.g., for other cancers) or is associated with certain changes in the bone marrow known as myelodysplasia [73]. The active substances in Vyxeos are daunorubicin and cytarabine in a 1:5 molar ratio, and they have been used together to treat leukaemia and other types of cancer for many years. They interfere in different ways with the production of new DNA within cells. The 1:5 ratio of the active ingredients has demonstrated maximum antitumor synergy in vitro and in vivo [74,75]. The liposomal formulation represents a posological improvement over the standard treatment and a simplification as both drugs are already mixed in the vial. Moreover, the liposomes of daunorubicin and cytarabine remain in the patient’s body for longer than conventional daunorubicin and cytarabine medicines and build up in the patient’s bone marrow. The liposomes protect the cancer drugs from being broken down early, which enhance their effect on cancer cells [73]. Liposomal Vyxeos improved survival compared with conventional daunorubicin and cytarabine in patients with acute myeloid leukaemia who have a poor prognosis and few alternatives. The side-effects were similar to the known side-effects of the active substances and were considered manageable [76].

On 24 March 2022, the Committee for Medicinal Products for Human Use adopted a positive opinion for the pegylated liposomal product **Zolsketil**, recommending the granting of a marketing authorization for the treatment of metastatic breast cancer, advanced ovarian cancer, progressive multiple myeloma, and AIDS-related Kaposi’s sarcoma. The active substance of Zolsketil is doxorubicin hydrochloride and will be available as a 2 mg/mL concentrate for solution for infusion. Studies have demonstrated its bioequivalence to Caelyx, which also contains doxorubicin hydrochloride in a pegylated liposomal formulation. At the moment, Zolsketil is pending EC decision (last updated 25 March 2022) and detailed recommendations for the use of this product will be published in the European public assessment report [77].

## 4. Clinical Trials with Nanoliposomes in Cancer Therapy

To find which nanoliposomal antineoplastic pharmaceuticals are undergoing Phase 4 studies, (ClinicalTrials.gov accessed on 8 March 2022) was used with the keywords liposome and cancer and the filter ‘Phase 4’. Out of the 34 results obtained, those with a date of study completion after the end of 2021 were selected. Among them, only the following two trials referred to combinations of antineoplastic drugs with nanoliposomes.

The efficacy and safety of treatment with Paclitaxel liposome and S-1 as first-line therapy in advanced pancreatic cancer patients aged between 18 and 75 years [78] is under investigation. This study is still in the preparation phase, and patient recruitment has not yet been completed.At the same stage is another study in adult women with epithelial ovarian cancer, to compare the efficacy of the combination of pegylated liposomal doxorubicin with carboplatin (experimental study group) versus the combination of paclitaxel with carboplatin (active comparator) [79].

The search for phase 3 trials on ClinicalTrials.gov (accessed on 8 March 2022), using the keywords liposome and cancer with the filter ‘Active, not recruiting’ and ‘with results’, gave 23 studies found. Of these, 14 did not use liposomes, although the word ‘liposome’ and cancer appeared in the corresponding file. The rest referred to clinical trials combining different antineoplastic agents or radiotherapy treatments with already marketed anticancer liposomal formulations, so they do not offer novelty within the subject under discussion. When the search was extended to studies in ‘recruiting’ and ‘Not yet recruiting’ status, 50 results were found, among which there were no new antineoplastic formulations with liposomes.

Among the results of the search in previous clinical phases, we can mention the following:Rhenium Nanoliposomes in recurrent glioma (ReSPECT) [80]. Rhenium 186 emits beta and gamma particles and has a half-life of 3.8 days. It can complex with hydroxyethylene diphosphonic acid to target it to bone. It has been studied in patients with metastatic cancers. It has shown both analgesic and therapeutic effects. In this clinical dual-phase 1/2 trial, its maximum tolerated dose, safety, and efficacy have been studied. Its completion is planned for 2025.C6 Ceramide NanoLiposome (CNL) in patients with relapsed/refractory acute myeloid leukemia [81] is an interesting phase 1 study to assess if combinations with this type of liposomes improve the clinical therapeutic index of already known antineoplastics in myeloid leukemia [82]. Completion is scheduled for June 2022.Liposomal SN-38 in treating patients with metastatic colorectal cancer [83]. SN-38 is a topoisomerase I inhibitor derivative of irinotecan that is about 1000 times more potent. In spite of this, the phase 2 trial ended in 2010 and concluded that this treatment did not improve survival rate [84].

Table 2 summarizes liposome-based products in clinical trials for cancer treatment. In summary, despite the growing number of basic research articles on the vehiculation of anticancer drugs by liposomes, more translational studies of these results to clinical reality are needed.

## 5. Conclusions and Future Prospects

Most conventional chemotherapeutics have shown limited clinical utility either by the incapacity to deliver active ingredients at therapeutic concentrations to the target tissues or by severe toxic effects on normal organs and tissues. Advancements in nanotechnology have contributed to the implementation of nanosystems in the cancer therapy for the improvement of treatment of metastatic tumors. It is expected that nanocarrier-based antineoplastic drugs’ delivery platforms will reach new heights in the coming years and will bring some significant changes in oncology research. In fact, the need for successful clinical translation of these nanocarriers is nowadays one of the main objectives, always trying to improve the quality of life in patients with cancer. Many drugs have been incorporated into liposomes to provide selective delivery to the target site. As oncology evolves toward developing more targeted therapies, anticancer drug delivery systems based on nanoliposomes have emerged as a promising candidate. Nevertheless, nanomedicine still has several limitations on its clinical potential, which is also related to safety. These limitations include, for instance, the lack of reproducibility of particle characteristics and load, where an appropriate scaling method is necessary, as the development and production of many liposome types often remain at laboratory-scale. In fact, small deviations in the production conditions may lead to changes in the nanosystem properties, with consequent alterations in the in vivo biodistribution, and therapeutic efficacy and/or safety. It is still necessary to provide a solid dosage form intended for different routes of administration (i.e., parenteral, oral, nasal, and/or pulmonary). Likewise, the composition of the liposome membrane can deeply influence the pharmacokinetics and tissue distribution of the drug. There should be focus on the clinical therapeutic effects and toxic side-effects of liposome lipid composition.

In recent years, the majority of clinical trials in cancer therapy have been based on the combination of conventional antineoplastic agents or radiotherapy treatments together with marketed anticancer liposomal products. Formulations of nanoliposomes containing new antineoplastic drugs should be desirable. The future focus of research could be directed to develop multifunctional targeted nanoliposomes using new antineoplastic drugs, different types of existing drugs, or new standardized methodologies easily translated into industrial scale.

## Figures and Tables

**Figure 1 ijms-23-04249-f001:**
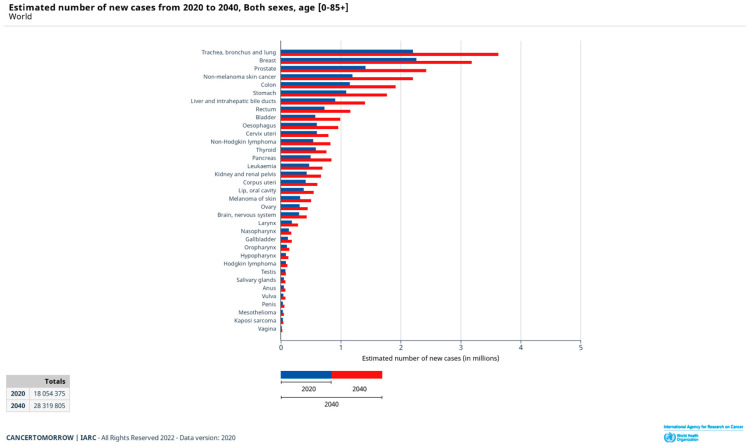
Estimated number of new cancer cases worldwide from 2020 to 2040 [1].

**Figure 2 ijms-23-04249-f002:**
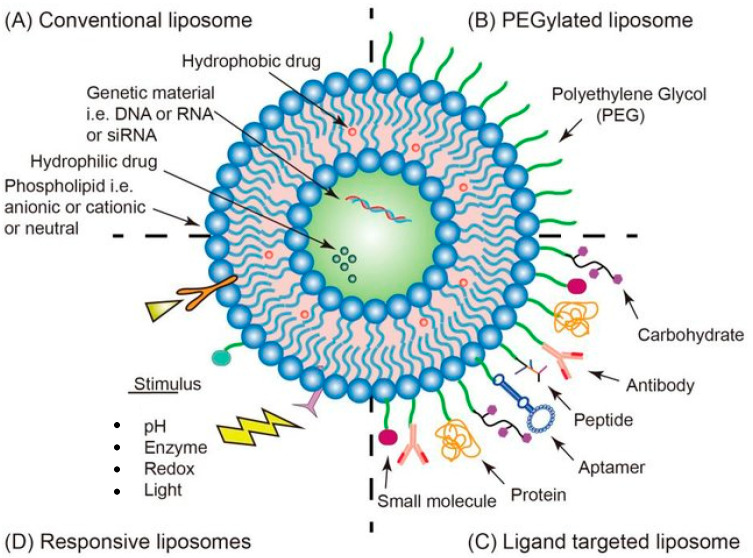
Strategies of functionalized liposome delivery for solid tumor therapy. (**A**) Conventional liposomes can reach tumor tissues through the EPR effect. (**B**) Surface-PEGylated liposomes can increase circulation time. (**C**) Functionalized liposomes modified with appropriate ligands can reach the target site and release the drug. (**D**) Responsive liposomes activate drug release only under specific environmental conditions. Reproduced from ref. [51].

**Figure 3 ijms-23-04249-f003:**
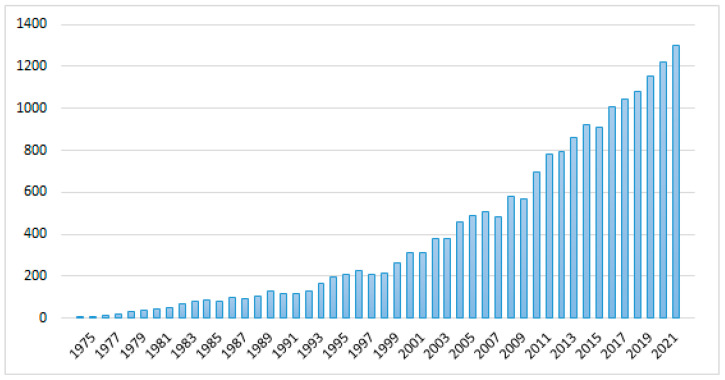
Evolution of the number of publications on (nano)liposomes and cancer.

**Table 1 ijms-23-04249-t001:** Liposome-based products marketed for cancer treatment.

Product Name	Active Ingredient	Approved Indication	Effective Dosage	Side-Effects (May Affect More Than 1 in 5 or 1 in 10 People)	Marketing Status	Regulatory Organism	Ref (Agency’s Website)
Marqibo	Vincristine sulphate	Acute lymphoid leukaemia	Every 7 days by infusion at a dose of 2.25 mg/m^2^	Constipation, nausea, pyrexia, fatigue, peripheral neuropathy, febrile neutropenia, diarrhea, anemia, decreased appetite, and insomnia	Discontinued	FDA	https://www.accessdata.fda.gov/drugsatfda_docs/nda/2012/202497Orig1s000SumR.pdf (accessed on 6 April 2022)
Doxil	Doxorubicin hydrochloride	Breast and ovarian, multiple myeloma and Kaposi’s sarcoma	Dose depends on the condition it is used for and is calculated on the basis of the patient’s weight and height	Asthenia, fatigue, fever, anorexia, nausea, vomiting, stomatitis, diarrhea, constipation, hand-foot syndrome, rash, neutropenia, thrombocytopenia, and anemia	Prescription	FDA	https://www.accessdata.fda.gov/drugsatfda_docs/summary_review/2008/050718se7-033_SUMR.pdf (accessed on 6 April 2022)
Caelyx pegylated liposomal	Doxorubicin hydrochloride	Breast and ovarian, multiple myeloma and Kaposi’s sarcoma.	Dose depends on the condition it is used for and is calculated on the basis of the patient’s weight and height	Side-effects depend on the type of cancer being treated. The most common is nausea, palmar-plantar erythrodysesthesia syndrome, vomiting, stomatitis, rash, weakness, low blood cell counts, loss of appetite, hair loss, tiredness, diarrhea, constipation, and mucositis	Prescription	EMA	ema.europa.eu/medicines/human/EPAR/caelyx (accessed on 6 April 2022)
Myocet liposomal	Doxorubicin hydrochloride	Breast neoplasms	Every 3 weeks by infusion. Dose is calculated on the basis of the woman’s weight and height	Neutropenic fever, infection, neutropenia, thrombocytopenia, anemia, leucopenia, loss of appetite, nausea, vomiting, stomatitis, mucositis, diarrhea, hair loss, weakness, fever, pain, and rigors (shaking chills)	Prescription	EMA	ema.europa.eu/medicines/human/EPAR/myocet-liposomal (accessed on 6 April 2022)
Onivyde pegylated liposomal	Irinotecan	Pancreatic neoplasms	Every 2 weeks by infusion together with fluorouracil and leucovorin. Dose is calculated on the basis of the patient’s weight and height	Diarrhea, nausea, vomiting, loss of appetite, neutropenia, tiredness, weakness, anemia, stomatitis, and fever Onivyde pegylated liposomal must not be given to patients who had a severe hypersensitivity reaction to irinotecan in the past and to breastfeeding women	Prescription	FDA EMA	https://www.accessdata.fda.gov/drugsatfda_docs/nda/2015/207793Orig1s000TOC.cfm ema.europa.eu/medicines/human/EPAR/onivyde-pegylated-liposomal (accessed on 6 April 2022)
Daunoxome	Daunorubicin citrate	Acute myeloid leukaemia, Kaposi’s sarcoma	Every 2 weeks by infusion Dose is calculated on the basis of the patient’s weight and height	Myelosuppression, cardiotoxicity, alopecia, neuropathy	Discontinued Prescription	FDA EMA	ema.europa.eu/en/medicines /human/orphan-designations/eu308585 ema.europa.eu/en/documents/psusa/daunorubicin-list-nationally-authorised-medicinal-products-psusa/00000936/201506_en.pdf (accessed on 6 April 2022)
Vyxeos liposomal	Cytarabine + daunorubicin hydrochloride	Acute myeloid leukaemia	On days 1, 3, and 5 of the first treatment course. Further courses on days 1 and 3 by infusion Dose is calculated using the patient’s height and weight	Hypersensitivity, febrile neutropenia, oedema, diarrhea, colitis, mucositis, tiredness, muscle and bone pain, belly pain, decreased appetite, cough, headache, chills, arrhythmias, fever, sleep disorders, and hypotension	Prescription	EMA	ema.europa.eu/medicines/human/epar/vyxeos-liposomal (accessed on 6 April 2022)
Zolsketil pegylated liposomal	doxorubicin hydrochloride	Breast and ovarian, multiple myeloma and Kaposi’s sarcoma			Initial authorization (positive opinion 24 March 2022)	EMA Pending EC decision	ema.europa.eu/en/medicines/human/summaries-opinion/zolsketil-pegylated-liposomal (accessed on 6 April 2022)

Abbreviations: FDA, Food and Drug Administration. EMA, European Medicine Agency.

**Table 2 ijms-23-04249-t002:** Liposome-based products in clinical trials for cancer treatment.

NCT Number	NCT04217096	NCT03794778	NCT01906385	NCT04716452	NCT00311610
Title of the study	Efficacy and safety of paclitaxel liposome and S-1 as first-line therapy in advanced pancreatic cancer patients	Evaluation of PLD Combined With Carboplatin Versus Paclitaxel Plus Carboplatin in the First-line Treatment of Epithelial Ovarian Cancer	Maximum Tolerated Dose, Safety, and Efficacy of Rhenium Nanoliposomes in Recurrent Glioma	Study of C6 Ceramide NanoLiposome in Patients With Relapsed/Refractory Acute Myeloid Leukemia	Liposomal SN-38 in Treating Patients With Metastatic Colorectal Cancer
Conditions	Advanced Pancreatic Cancer	Efficacy and Safety	Glioma	Acute Myeloid Leukemia, in Relapse|Acute Myeloid Leukemia, Refractory	Colorectal Cancer
Interventions	Drug: Paclitaxel liposome|Drug: S-1	Drug: pegylated liposomal doxorubicin|Drug: paclitaxel|Drug: Carboplatin	Drug: Rhenium Liposome Treatment	Drug: Ceramide NanoLiposome (Ceraxa)	Drug: SN-38 liposome
Outcome Measures	Progression free survival|Overall Response Rate|overall survival|Disease control rate|Quality of life|Adverse events	PFS|OS|ORR|DCR|the incidence and severity of adverse reactions|quality of life assessment	Maximum Tolerated Dose|Dose Distribution|Response rate|Survival	Number of Patients with Dose-Limiting Toxicities, and with Adverse Events|Severity and Duration of Adverse Events|Dose Levels achieved during study|Half Life|Clearance|Other	Objective response rate|Toxicity|Progression-free survival|Overall survival
Gender	All	Female	All	All	All
Age	18 Years to 75 Years (Adult, Older Adult)	18 Years to 75 Years (Adult, Older Adult)	18 Years and older (Adult, Older Adult)	18 Years and older (Adult, Older Adult)	18 Years to 120 Years (Adult, Older Adult)
Phase	4	4	1|2	1	2
Enrolment	40	396	55	18	30
Sponsor	Fudan University	Women’s Hospital School Of Medicine Zhejiang University	Plus Therapeutics|National Cancer Institute (NCI)	Keystone Nano, Inc.|University of Virginia|Memorial Sloan Kettering Cancer Center|Milton S. Hershey Medical Center	Alliance for Clinical Trials in Oncology|National Cancer Institute (NCI)
Study Designs	Allocation: N/A|Intervention Model: Single Group Assignment|Masking: None (Open Label)|Primary Purpose: Treatment	Allocation: Randomized|Intervention Model: Parallel Assignment|Masking: None (Open Label)|Primary Purpose: Treatment	Allocation: N/A|Intervention Model: Single Group Assignment|Masking: None (Open Label)|Primary Purpose: Treatment	Allocation: N/A|Intervention Model: Single Group Assignment|Masking: None (Open Label)|Primary Purpose: Treatment	Allocation: N/A|Intervention Model: Single Group Assignment|Masking: None (Open Label)|Primary Purpose: Treatment

Abbreviations: PLD, pegylated liposomal doxorubicin. PFS, Progression-free survival. OS, Overall survival. ORR, objective response rate. DCR, Disease control rate.

## Data Availability

Not applicable.

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
