# Peer review of "Nanoliposomes in Cancer Therapy: Marketed Products and Current Clinical Trials"

_ijms, 2022, doi:10.3390/ijms23084249_

Round 1

Reviewer 1 Report

I would request a minor revision, with the following questions:

  1. In Section 2nd, the authors discuss the phospholipid bilayers composition of the liposomes (“Phospholipids with or without incorporation of cholesterol, surfactants and other materials are used for their preparation.”), and the consequent changes in their properties, considering the different proportions of saturated and/or unsaturated phospholipids.
    I would suggest adding a sentence on the presence of cholesterol at different levels, due to its role in the fragility/stiffness of the membrane.
  1. “..with the help of certain molecules such as opsonins” (line 154): I would suggest specifying “of certain host molecules such …”.
  2. In Section 3rd (line 206), “a process that involves a series of biochemical changes”: I would suggest that changes in the malignant phenotype as a whole (not just biochemical) have been also considered.
  3. Among the nanoliposome applications, the field of therapy cancer stem cell targeting could be mentioned. Could the authors add some things about this?
  4. In Figure 2, at the left, below, align the terms with their points.

Author Response

We thank the reviewers for their valuable comments and suggestions. In the following, we cite the reviewer’s revision answering point by point to their comments and requirements:

  1. In Section 2nd, the authors discuss the phospholipid bilayers composition of the liposomes (“Phospholipids with or without incorporation of cholesterol, surfactants and other materials are used for their preparation.”), and the consequent changes in their properties, considering the different proportions of saturated and/or unsaturated phospholipids. I would suggest adding a sentence on the presence of cholesterol at different levels, due to its role in the fragility/stiffness of the membrane.

According to reviewer’s suggestion a new sentence referred to the presence of cholesterol has been added (see line 120): “In this sense, presence of cholesterol enhances the mechanical strength of membranes, influencing membrane elasticity, and increasing the packing density of lipids as function of the proportion included. Moreover, with the incorporation of cholesterol, the liposomes prepared are expected to be more stable in presence of biological fluids (Chem Phys Lipids. 1993 Sep;64(1-3):275-85. doi: 10.1016/0009-3084(93)90071-a). Due to this, cholesterol is often incorporated in the formulation of liposomes used in drug delivery (Sci Rep. 2014 May 21;4:5005. doi: 10.1038/srep05005)”.

  1. “..with the help of certain molecules such as opsonins” (line 154): I would suggest specifying “of certain host molecules such …”.

This sentence has been adapted according to reviewer’s comment (line 166).

  1. In Section 3rd (line 206), “a process that involves a series of biochemical changes”: I would suggest that changes in the malignant phenotype as a whole (not just biochemical) have been also considered.

We agree with reviewer that changes in the malignant phenotype need to be considered as a whole, not only biochemical. This sentence has been modified (line 217).

  1. Among the nanoliposome applications, the field of therapy cancer stem cell targeting could be mentioned. Could the authors add some things about this?

We appreciate the comment of the reviewer regarding the important role of CSCs in cancer therapy. A new paragraph has been added (line 244): “In this line of action, cancer stem cells (CSCs) are a subject of intense research that aims the development of efficient targeting therapies. CSCs have been identified in almost all types of cancers and can act as a reservoir of cancer cells that may cause a relapse after surgery, radiation, or chemotherapy. In fact, chemoresistance is a major problem in cancer therapy that contribute to poor prognosis and survival rates of pa-tients. CSCs play a key role in this event due to their unlimited proliferative ability and multidrug resistance. (Cells. 2019 Aug 18;8(8):926. doi: 10.3390/cells8080926). Thus, they are obvious targets in therapeutic approaches and also a great challenge in cancer treatment (Front Pharmacol. 2017; 8: 1. doi: 10.3389/fphar.2017.00001). In this sense,liposome nanoparticles, with various sizes and modifications to their surfaces can be easily prepared and offer promising means for targeting and eradicating of CSCs for cancer therapies (ACS Biomater Sci Eng. 2021 Jun 14;7(6):2508-2519. doi: 10.1021/acsbiomaterials.1c00110).”

        5. In Figure 2, at the left, below, align the terms with their points.

Figure 2 comes from reference “Med. Drug Discovery 2020, 6, 100024. doi: 10.1016/j.medidd.2020.100024”. However, trying to improve it, and according to reviewer’s opinion we have insert some points to align them with the text (with track changes).

Reviewer 2 Report

Talens-Visconti et al. have reviewed on application of nanoliposomes in cancer therapy focusing mainly on marketed products and current clinical trials. Although it is an interesting piece of work, the presentation should be substantially improved to contain some more in-depth technical details. Therefore I recommend authors to carefully revise in accordance with the following comments before this paper could be accepted for publication:

  1. Title – the title is more objective and it should be modified as “Nanoliposomes in cancer therapy: Marketed products and current clinical trials” or something similar to this.
  2. Abstract – It is too general. The authors should rewrite the abstract based on the content outlined and described in this review. Most importantly, it should contain the highlights of key findings and correlations based on the articles reviewed in this paper.
  3. I do not understand the need for splitting into too many one/two sentence paragraphs. The authors should combine many small paragraphs to have few large paragraphs.
  4. Figure 1 – the labels are too small to aid readability. They should be enlarged for clarity.
  5. Table 1 – some reference citations should be provided for each product. Also, the details of effective dosage and side effects should be included.
  6. A new Table 2 should be included to tabulate the current clinical trials described under section 4.
  7. Section 4 – some more technical protocol/data/findings on each nanoliposome clinical trial should add more interest to readers.
  8. L287 – “Onivyde” should be formatted as bold.
  9. Conclusion – the existing limitations/bottlenecks and research gaps should be clearly identified for the readers to have quick grasp of future perspective on the topic.

Author Response

We thank the reviewers for their valuable comments and suggestions. In the following, we cite the reviewer’s revision, answering point by point to the comments and requirements, and clarifying the changes introduced in the manuscript in order to improve our work and reach the publishable version.

  1. Title – the title is more objective and it should be modified as “Nanoliposomes in cancer therapy: Marketed products and current clinical trials” or something similar to this.

According to reviewer’s opinion we have modified the title of the manuscript as suggested.

  1. Abstract – It is too general. The authors should rewrite the abstract based on . the content outlined and described in this review. Most importantly, it should contain the highlights of key findings and correlations based on the articles reviewed in this paper.

We agree with the reviewer and in consequence the abstract has been rewritten highlighting the key findings of the manuscript:

“The drugs currently used for cancer treatment have many drawbacks, since they damage both tumor cells and healthy cells and, in addition, they tend to be poorly soluble drugs. Their transport in nanoparticles can solve these problems since these can release the drug into tumor tissues, as well as improve their solubility, bioavailability and efficacy, reducing their adverse effects. This article focuses on the enormous advantages that nanotechnology can bring to medicine, with special emphasis on nanoliposomes. For this purpose, a review has been made of the nanoliposomal sys-tems already marketed for the treatment of cancer, as well as those that are in the research phase, highlighting the clinical trials that are being carried out. All marketed liposomes studied are in-travenously administered showing a reduced intensity of side effects compared with the non-liposomal form. Doxorubicin is the active ingredient most frequently employed. Ongoing clinical trials expand the availability of liposomal medicines with new clinical indications. As on-cology evolves toward developing more targeted therapies, anticancer drug delivery systems based on nanoliposomes have emerged as a promising candidate. In conclusion, the introduction of drugs in nanoliposomes means an improvement in their efficacy and in the quality of life of patients. The future focus of research could be directed to develop multifunctional targeted nanoliposomes using new anticancer drugs, different types of existing drugs or new standardized methodology easily translated into industrial scale.”

  1. I do not understand the need for splitting into too many one/two sentence paragraphs. The authors should combine many small paragraphs to have few large paragraphs.

Authors are in accordance with reviewer’s comment about the possibility of combine small paragraphs and we have tried to improve the reading of the manuscript. For example, see lines 142, 171, 185, 241.

  1. Figure 1 – the labels are too small to aid readability. They should be enlarged for clarity.

Figure 1 comes from GLOBLOCAN website (https://gco.iarc.fr/), therefore it is automatically drawn when different parameters are selected. There is no way to increase the labels’ size if we do not eliminate some types of cancers to improve the readability. However, if reviewer consider it appropriate, we can prepare a new figure including less types of cancer.

  1. Table 1 – some reference citations should be provided for each product. Also, the details of effective dosage and side effects should be included.

According to reviewer’s comments a new table has been prepared. In this new version more information, such as effective dosage and the most common side effects for each product, have been included. References to EMA or FDA websites, where readers can find additional information, have also been added. Lastly, a new pegylated liposomal product (Zolsketil) has been considered as it has obtained initial authorisation on 24-03-2022. A new paragraph has been incorporated in the manuscript regarding this product (line 368):

“On 24 March 2022, the Committee for Medicinal Products for Human Use adopt-ed a positive opinion for the product Zolsketil pegylated liposomal recommending the granting of a marketing authorisation for the treatment of metastatic breast cancer, advanced ovarian cancer, progressive multiple myeloma and AIDS-related Kaposi's sarcoma. The active substance of Zolsketil is doxorubicin hydrochloride and will be available as 2 mg/ml concentrate for solution for infusion. Studies have demonstrated its bioequivalence to Caelyx, which also contains doxorubicin hydrochloride in a pegylated liposomal formulation. At the moment, Zolsketil is pending EC decision (last updated 25-03-2022) and detailed recommendations for the use of this product will be published in the European public assessment report (https://www.ema.europa.eu/en/ docu-ments/smop-initial/chmp-summary-opinion-zolsketil-pegylated-liposomal_en.pdf).”

  1. A new Table 2 should be included to tabulate the current clinical trials described under section 4.

A new table, 2, has been incorporated to the manuscript summarizing the current clinical trials.

  1. Section 4 – some more technical protocol/data/findings on each nanoliposome clinical trial should add more interest to readers.

The table 2, that has been added to the manuscript, incorporates the requested information.

  1. L287 – “Onivyde” should be formatted as bold.

Onivyde has now been marked in bold (line 315).

  1. Conclusion – the existing limitations/bottlenecks and research gaps should be clearly identified for the readers to have quick grasp of future perspective on the topic.

Following reviewer’s comments relate to conclusion, we have improved this section trying to clarify the existing limitations and research gaps to help readers get a clear idea about the future perspective on the topic of the manuscript. See line 443:

“Nevertheless, nanomedicine still has several limitations on its clinical potential, which is also related to safety. These limitations include, for instance, the lack of reproducibility of particle characteristics and load, being necessary an appropriate scaling method, as the development and production of many liposome types often remain at laboratory-scale. In fact, small deviations in the production conditions may lead to changes in the nanosystem properties, with consequent alteration of the in vivo bio-distribution, and therapeutic efficacy and/or safety. It is still necessary to provide a solid dosage form intended for different routes of administration (i.e., parenteral, oral, nasal and/or pulmonary). Likewise, the composition of the liposome membrane that can deeply influence the pharmacokinetics and tissue distribution of the drug. There should be focus on the clinical therapeutic effects and toxic side effects of liposome li-pid composition.”

Round 2

Reviewer 2 Report

The authors have satisfactorily addressed all the comments raised by the reviewers and therefore I recommend acceptance of this article for publication in IJMS.